# Exploring the Synergy of Music and Medicine in Healthcare: Expert Insights into the Curative and Societal Role of the Relationship between Music and Medicine

**DOI:** 10.3390/ijerph20146386

**Published:** 2023-07-18

**Authors:** Juliane Hennenberg, Manfred Hecking, Fritz Sterz, Simeon Hassemer, Ulrich Kropiunigg, Sebastian Debus, Kurt Stastka, Henriette Löffler-Stastka

**Affiliations:** 1Department of Biomedical Imaging and Image-Guided Therapy, Medical University of Vienna, 1090 Wien, Austria; juliane.hennenberg@meduniwien.ac.at; 2Department of Internal Medicine III, Medical University of Vienna, 1090 Wien, Austria; manfred.hecking@meduniwien.ac.at; 3Department of Emergency Medicine, Medical University of Vienna, 1090 Wien, Austria; fritz.sterz@meduniwien.ac.at; 4Department of Psychoanalysis and Psychotherapy, Medical University of Vienna, 1090 Wien, Austria; sdfhassemer@gmail.com; 5Department of Medical Psychology, Medical University of Vienna, 1090 Wien, Austria; ulrich.kropiunigg@meduniwien.ac.at; 6Department of Vascular Medicine, Vascular Surgery-Angiology-Endovascular Therapy, University Medical Center of Hamburg-Eppendorf, 20251 Hamburg, Germany; s.debus@uke.de; 7Department of Psychiatry, Klinik Favoriten Hospital of Vienna, 1100 Wien, Austria; kurt.stastka@gesundheitsverbund.at; 8Mental Health and Behavioural Medicine Program, Medical University of Vienna, 1090 Wien, Austria

**Keywords:** prevention, identification, awareness, curative function, treatment, societal role, non-verbal relationship, creativity

## Abstract

Our study aimed to investigate the correlation between medicine, health perception, and music as well as the role of music in the healthcare setting. To gain insights into the dynamics between these two fields, we gathered opinions from attendees and presenters at an international conference on music medicine, musicians’ health, and music therapy. A team of six interviewers conducted a total of 26 semi-structured interviews. The interview guide focused on four predetermined themes: (1) “music in medicine”, (2) “performing arts medicine”, (3) “music for the individual”, and (4) “music for society”. The responses were analyzed using grounded theory methods as well as thematic and content analysis. To enhance the analytical strength, investigator triangulation was employed. Within the predefined themes, we identified several subthemes. Theme 1 encompassed topics such as “listening and performing music for treating diseases and establishing non-verbal relationships”, “the value of music in specific disorders, end-of-life care, and pain management”, and “the design of sound spaces”. Theme 2 explored aspects including the “denial and taboo surrounding physical and mental health issues among musicians”, “the importance of prevention”, and an antithesis: “pain and suffering driving creativity”. Theme 3 addressed the “mental role of music in ordinary and extraordinary life” as well as “music’s ability to enable self-conditioning”. Lastly, Theme 4 examined the role of music in “cultural self-identification” and “development and education for children”. Throughout the interviews, participants expressed a lack of knowledge and awareness regarding interdisciplinary research and the fields of music and medicine. Our findings affirm the significance of music therapy and performing arts medicine as well as the broader relationship between music and medicine. They highlight the potential benefits of perception and experiential pathways for individuals and, consequently, for human society.

## 1. Introduction

Throughout the history of medicine, music has played a prominent and intertwined role. For instance, the Assyrians in 2000 BC documented the use of music to counteract the influence of evil spirits [1]. In ancient Greece, Apollo was venerated as the god of both music and healing [2]. Plato eloquently expounded on the salutary effects of music on health in his Socratic dialogue “Politeia” [3]. Additionally, the Old Testament recounts King Saul finding solace from an “evil spirit” through the soothing sounds of David’s lyre [4]. Music’s significance persisted through the Middle Ages, as it formed a fundamental part of medical education in Europe [5]. In the late 19th century, recognition of music’s therapeutic potential led to extensive research endeavors [6] and its integration into medical practice [7].

This deep integration of music within medicine gave rise to performing arts medicine, an occupational medicine discipline that addresses the etiology, treatment, and prevention of health issues encountered by performing artists [8]. As early as the 18th century, the physiological impact of music-making was observed, with 19th-century publications detailing injuries among keyboard and violin players as well as the occurrence of musician’s cramp (focal dystonia) [9]. Notably, performing arts medicine extends beyond empathy, as music contributed a substantial 81.9 billion EUR to the European Union’s gross domestic product in 2018 [10].

Moreover, the connection between music and medicine transcends mere curative aspects, as medicine can permeate the realm of music through the lens of health perception for the listener. The experience of music from performers or composers who may exhibit health problems or push the boundaries of their physical well-being differs from attending a performance where musicians appear to be in good health. Throughout the centuries and across genres, numerous renowned artists have openly shared their struggles with health issues and mental challenges while projecting an image of perfect health and ideal aging during their performances, an aspect intricately interwoven with the overall impact of their artistry.

As authors of this analysis, we acknowledge that we do not claim expertise in this relationship solely based on formal training. Rather, our objective was to deepen our understanding of this connection, exploring the potential value of music in medicine and the reciprocal influence of medicine and overall health on music creation and experiences. To achieve this, we conducted a qualitative interview study at the “Science and Sounds” international conference, dedicated to “music medicine, musicians’ health, and music therapy”. Our aim was to utilize a thematic analysis of expert opinions to address the current knowledge gap regarding the intricate relationship between music and medicine, while identifying potential areas for future research and funding.

## 2. Materials and Methods

### 2.1. Conference Details and Study Participants

The “Science and Sounds” international conference was held 8–10 September 2022, in Hamburg, Germany [11]. It was a joint project between the University of Music and Drama, Hamburg, and the University Medical Center, Hamburg-Eppendorf, Germany, and was conducted in collaboration with the International Society for Music in Medicine [12]. The main presentations were given by international experts from the fields of musicians’ health, music medicine, and music therapy. The main focus of the conference was on the domain of music medicine and music therapy, with prominent international experts in the field [13]. Invited presenters and attendants of this conference were randomly invited (i.e., without specific selection criteria) to be interviewed for the present study. No repeat interviews were conducted.

### 2.2. Ethical Considerations, Approval, and Consent to Participate

This study was carried out in accordance with relevant national and international guidelines and regulations. The Ethics Committee of the Medical University of Vienna determined that ethics approval was not required according to national regulations, and due to the fact that the relationship between interviewers and interviewees was one of peers rather than, for example, a relationship involving patient-doctor dependence (communication with the Ethics Committee will be made available upon request). All study participants gave their written informed consent to participate in this study, and all interviews were anonymized before the analysis.

### 2.3. Investigators

The interviewers comprised six co-authors of the present article (J.H., S.H., K.S., H.L.-S., F.S. and M.H.). Members of this team ranged from 26 to 66 years old, and included four men and two women (self-declared), who conducted all but one interview on a one-to-one basis with the interviewees.

### 2.4. Interview Setting

The “Science and Sounds” conference took place at two venues: the University of Music and Drama, Hamburg [14], one of the larger universities of music in Germany; and the University Medical Center, Hamburg-Eppendorf. Semi-structured face-to-face single and group interviews were predominantly scheduled during the intermissions of the “Science and Sounds” conference, and were conducted at the University of Music and Drama and the University Clinic. In specific, group interviews in this setting allowed for the exploration of shared perspectives, facilitated dynamic discussions, and stimulated collective thinking. However, the presence of a group setting may have introduced social influence or conformity, potentially influencing participants’ responses and limiting the expression of individual viewpoints. None of the investigators had previous familiarity with the meeting venue, and the interviews were conducted in the most convenient remote areas of the venue building.

### 2.5. Interview Guide

A semi-structured interview guide (see Appendix B) was developed with consensus of the researchers. Specifically, authors SH and MH made independent suggestions, which were discussed, adopted, and finalized among the group of six interviewers. To address potential bias stemming from rigid interview structures, participants agreed to employ a flexible interview outline that could be adapted based on their responses. This approach aimed to promote a more organic and nuanced exploration of the topic. Additionally, establishing rapport and considering diverse perspectives within the research team helped ensure a comprehensive and unbiased analysis of the collected data. German was the preferred interview language, although English was used when necessary. Investigators initially introduced themselves to the participants, and gave a short explanation of the research objectives. After asking the participants for some central demographic information, the main aim was to explore possible interdependencies of music and medicine, in four pre-specified themes: (1) “music in medicine”, (2) “performing arts medicine”, (3) “music for the individual”, and (4) “music for society”. Participants were encouraged to openly express their views, and anonymity was guaranteed in all aspects. To gain a comprehensive understanding, valuable insights of the interviewees’ personal experiences were obtained by asking, for example, about their personal limitations. This was conducted with full respect for the interviewees’ autonomy and comfort, as disclosure of such limitations during the interviews was voluntary and not compulsory. The decision to forgo the typical development of the semi-structured interview guide within focus groups was driven by participant availability, and a desire to capture diverse individual perspectives more comprehensively, and individual interviews were chosen as an alternative approach to ensure efficient data collection and optimize the depth of understanding.

### 2.6. Recruitment Goal, Data Transcription and Translation, and Analysis

Our goal was to recruit up to 30 participants, depending on how soon data saturation would be reached. The interviewers discussed whether saturation had been reached three times between the interview rounds. At the third discussion time-points, when 26 interviews had been completed, the co-authors (particularly J.H., H.L.-S., and F.S.) voiced that the responses they were obtaining were increasingly overlapping and that it would be sensible to refrain from recruiting more interviewees. No specific person was assigned to recruitment. Instead, the congress president was asked to announce, introduce, and spread word-of-mouth information of our plans at the beginning of the conference sessions. No willing participant was excluded. Interviews were audio-recorded and transcribed using basic standard transcription systems supplying dialogue lists and documents [15]. The study investigators performed translation into English.

While striving for data saturation and diverse perspectives, the recruitment goal of up to 30 participants may have introduced complexities that necessitate meticulous attention, such as the possibility of variations in participant selection and representation. It is crucial to acknowledge and address these factors in the discussion section, as they can potentially influence the interpretation of the findings and the generalizability of the study results.

Seeking data saturation was essential to ensure a comprehensive understanding of the research topic, as it signifies that no new information or themes were emerging from additional interviews or data sources, enhancing the credibility and robustness of the study findings.

### 2.7. Data Analysis

The analytical coding process used in this study followed a mixed method approach, applying procedural rules from grounded theory [16,17], as well as principles from qualitative content and thematic analysis [18,19]. After author UK held an instruction session, the anonymized interview transcripts were thoroughly coded by authors J.H., F.S., and M.H., and were then assigned to subthemes within four pre-specified themes using a text-sorting technique (TST) for qualitative data analysis [20], based on the Microsoft Word program. Authors J.H., F.S., and M.H. worked individually on these subthemes and arranged four joint coding sessions in which they verified that all subthemes represented essential patterns and relationships within the data set. Next, two coder triangulation sessions were arranged with author UK, as well as a support team triangulation session with authors S.H., H.L.-S. and S.D. The Consolidated Criteria for Reporting Qualitative Research (COREQ) was utilized to enhance the transparency, rigor, and comprehensive reporting of qualitative research methods and findings in this study [21].

## 3. Results

### Descriptive Results

Here, we present the results from 26 interviews conducted with participants of the “Science and Sounds” international conference held in Germany in September 2022 [22]. In one-to-one interviews, we asked open questions. The responses revealed that additional resources are needed to raise awareness for and to support music therapy as well as to address musicians’ medical needs. Analysis of the responses also revealed issues related to the use of music in medicine and, vice versa, to diseases related to performing music.

Table 1 shows the demographics and personal characteristics of the 26 interviewed study participants. Among the participants, 18 self-identified as women and 6 as men; 23 resided in Germany; 22 stated they were of Caucasian ethnicity; and 20 indicated that their native language was German. Most of the participants (*n* = 10) were 21–30 years of age. While 4 interviewees did not self-identify as musicians, 6 reported that they spent >10 h per week with active music making, 13 reported 2–10 h per week, and 3 reported <2 h per week. Six interviewees worked as music therapists, six were students or in training, and four were medical professionals (see Table 1 for further details). The collection of participants was made in an effort to encompass a diverse cohort and obtain a range of perspectives.

Table A1 (see Appendix A) presents the themes and subthemes. The four themes were pre-specified based on our interview guide (see Section 2 and Appendix B). Our thematic analysis (carried out by authors J.H., F.S. and M.H.) revealed up to six subthemes for each theme. Some of the identified subthemes re-appeared throughout the four themes. For example, participants expressed that they experienced a popular lack of knowledge and awareness of the impact of music in the medical field (e.g., within theme 1, e.g., subtheme 6: They are unaware that music therapy exists) and of the importance of medicine for music and musicians (within theme 2, e.g., subtheme 2: I still find it frightening how other instrumentalists, in particular, treat their bodies). Likewise, subtheme 2 from theme 3 (“mental role of music in ordinary and extraordinary life”) was somewhat similar to subtheme 2 from theme 4 (“music for cultural self-identification”) and subtheme 5 from theme 1 (“overcoming loneliness”).

When we further condensed all subthemes from the four pre-specified themes, we found that music itself was perceived as a common language outside of human speech, rooted in childhood, with importance for cultural self-identification, individual development, and education. Participants emphasized that music reduces stress and enables self-conditioning, albeit with variations in form and activity (active to passive). Two participants stated that music had to be “politically correct” (subtheme 5, theme 4). Interviewees also stated that the act of listening and/or performing aids in medical treatment, establishes a non-verbal relationship with the caretaker, can be applied in various (physical) environments (e.g., see subtheme 4, theme 1), and has special importance at the end of life. Based on their experience, interviewees stated that taking advantage of medical treatment was a taboo for most musicians. Nevertheless, the role of medicine for performers was considered important for the prevention of serious illness, while requiring interaction between multiple medical disciplines. As an interesting antithesis, at least two participants stated that medical treatment ran the risk of counteracting artistic development, based on the notion that creativity came from suffering (subtheme 5, theme 2: The absence of medicine as a creative boost; Illness and, above all, pain makes the “I” awaken to itself. Pain is a great awareness maker. Or also the crisis. In this respect, it is also a motor for creativity, without there being an inevitable link between disorder and artistic creation. It is a drive.).

## 4. Discussion

The findings were divided into four pre-specified themes, based on our interview guide (see Appendix B), which will be discussed in greater detail below. Within those four themes, we further identified 21 subthemes. All things considered, our interviewees voiced that there is an enormous demand for physicians and musicians to make more and better use of their corporate influence, not only for their own benefit but also to promote further growth of society in the field of music-medicine.

### 4.1. Theme 1: Music in Medicine

Regarding the first pre-specified theme, “music in medicine”, our study participants unanimously stated that both listening to music and performing music in a medical setting can benefit a patient’s progress in treatment (see Table A1, subthemes 1–2, theme 1). The act of making music together was described as helpful for establishing a personal relationship between a patient and their therapist or physician—enabling patients to “open up”, and fostering their emotional stability with different perceptions through experiential learning and new learning pathways. Interviewees also voiced that a “pleasant environment” supports the development of the patient’s relationship with a therapist and/or caretaker (see Table A1, subtheme 1, theme 1). These opinion-based subthemes are supported by research findings. Similar to rewards that activate the striatal dopaminergic system, intangible stimuli, such as music, can cause euphoria, cravings, and emotional arousal [23,24,25,26,27,28]. Such processes can be measured with functional magnetic resonance imaging, and do not depend on the individual having prior musical training [24,25,27].

During their interviews, our participants mentioned various specific medical fields in which the presence of music was particularly helpful in patients’ recuperation (see Table A1, subtheme 2–5, theme 1): […] Pregnant women, women who may have had a miscarriage and are now afraid to get pregnant again or who already are, premature babies, children in orthopedics, in oncology, patients with eating disorders, people in the very last phase of life, in palliative care and until the day they die. Overall, music intervention was said to be associated with an improvement in patients’ health-related quality of life (see Table A1, subthemes 1–3, theme 1). These statements are in line with numerous research studies showing that music intervention in various medical settings is a safe and affordable method of improving the course of a disease. Studies have shown that music improves patients’ physical and psychological outcomes during treatment, e.g., for hemodialysis [29,30,31,32,33]; in the hospice setting [34,35,36,37]; in the fields of neurology [23,38,39,40,41,42], psychiatry [43,44], and oncology [45,46,47,48]; during pregnancy [49]; or in COPD [50,51,52].

Also related to theme 1, interviewees often reported the impact of music on pain management: I have to mention pain as the area with the best study record (see Table A1, subtheme 2, theme 1). The literature includes many studies of music in perioperative settings for reducing anxiety and pain [53,54,55,56,57,58,59,60,61,62] as well as during invasive procedures in patients with coronary heart disease, where music has positively affected blood pressure, respiration rate, anxiety, pain level, and the amount of sedative medication required [63]. Another study showed that music reduced the need for benzodiazepines during cataract surgery [64]. Although the medical pain pathway includes the limbic system and is linked to emotion, it remains unknown exactly how music affects the pathway activation and, therefore, to what extent music alters pain reception [65].

Participants emphasized the impact of music, not only on patients but also on employees in the medical environment, such as physicians, especially surgeons listening to their favorite music during an intervention (see Table A1, subthemes 3 & 4, theme 1). A previous study showed that playing music during surgery improved intervention time, accuracy, and precision during laparoscopic procedures [66,67]. According to the literature, it seems important not to select background music that is too strongly liked or disliked, as this might also negatively affect concentration [68]. Nevertheless, in line with another previous study [69], our participants voiced that music exposure could enhance human cognitive performance (see Table A1, subtheme 3, theme 1): That this is good for me when operating. Specifically, I notice that with classical music I can concentrate much better in the manual-practical activity and then also dissect much more targeted and quickly.

The last notable aspect related to theme 1 is that interviewees reported positive impacts of music played in different locations of the medical environment, not only preoperatively but also in a hospital’s entrance hall or in the outpatient clinic. In such environments, music is considered to reduce stress and anxiety. Our interviewees most frequently mentioned classical music: When you bring music in as a feel-good factor in a hospital atmosphere that can be rather sterile and scary, you can break down a lot in patients up front. In line with subtheme 4 of theme 1 (see Table A1), previous research has shown that creating soundscapes can improve the experience of the hospital environment for both patients and providers [70]. One interviewee claimed that the use of compositions explicitly written for the entrance hall of a hospital created a comfortable and welcoming place for patients, and that they had already implemented this strategy in one of their city hospitals (see Table A1, subtheme 3, theme 1).

### 4.2. Theme 2: Performing Arts Medicine

The predefined Theme 2, “performing arts medicine”, encompassed the impact of medicine in the world of performing arts. The most frequent issue voiced by interviewees was the importance of advertising and promoting the use of targeted performing arts medicine (see Table A1, subtheme 2, theme 2). It was also emphasized that there are various different diseases that affect musicians. Interviewees stated that musicians need medical support for their daily life that is specific to their art. They also suggested that medical professionals and musicians could improve their collaboration by developing interdisciplinary “boards” for particular diseases, such as those that exist in oncology. Participants proposed that such a collaboration should also involve music teachers, therapists, and those research professionals (see Table A1, subtheme 4, theme 2). These results and subthemes are interesting in light of the achievements that have already been made in performing arts medicine. Performing arts clinics have now been established around the world, and collaboration exists between medical professionals with expertise in treating musicians, including orthopedics; neurology; ear, nose, and throat medicine; and psychology [71]. Likewise, research is being conducted and presented at meetings to further develop modern and preventative approaches. Our results might indicate that these efforts and studies are currently being ignored. It has previously been shown that there is a great need to advertise and promote performing arts medicine [72], and interdisciplinary interaction in performing arts medicine [73].

Albrecht Lahme stated “No overuse without misuse”, meaning that the severity of medical conditions among musicians is linked to and caused by the amount of practicing hours [74]. Our participants actually made several statements referring to overuse syndrome (Table A1, subtheme 2, theme 2), which is common among professional musicians [75]. Musicians’ intense focus on their instruments may affect their physical and mental health [76,77,78]. For example, enduring in an non-ergonomic position may result in musculoskeletal diseases [79,80,81]. Such musculoskeletal injuries may eventually necessitate medical consultation for treatment [82,83,84]. Similar opinions were given throughout our interviews: The circumstances. I still find it terrifying how other instrumentalists, in particular, deal with their bodies, the suffering they go through, and what they believe is necessary to reach a certain level. How the lack of education at the universities also guarantees that no awareness is developing.

It has previously been expressed that “a sixteenth note of prevention is worth a whole note of cure” [85]. Interviewees clearly felt that there was a serious need for preventive measures benefiting musicians. Importantly, the participants also questioned, for example, why screenings and check-ups are not yet being promoted for musicians, as there is a high demand [86] (see Table A1, subtheme 6, theme 1 and subtheme 4, theme 2).

The final topic identified in theme 2 is that the interviewees voiced that there is a need to improve the present lack of knowledge about prospects for improving musicians’ physical health (see Table A1, subthemes 1 and 2, theme 2). Interviewees also felt that psychological support must be offered to a greater extent, such as when dealing with performance anxiety, which interviewees felt could result in physical disorders (see Table A1, subtheme 2, theme 2). These perceptions are important because psychological support for performance anxiety may be among the best-researched areas in performing arts medicine, as indicated by the number of reviews and meta-analyses on this subject [87,88,89,90,91,92,93]. However, the interviewees’ opinions on this topic indicated that stage fright was not adequately being managed, despite the available therapeutic options.

### 4.3. Theme 3: Music for the Individual

The majority of interviewees discussed the importance of music in their everyday lives. Most participants had experienced the presence of music in their own childhood starting from a very early age (see Table 1) and explained that music had become an essential part of their personal lives growing up, giving them joy and stability. Herbert von Karajan and Daniel Barenboim are perhaps among the most famous musicians who have claimed that playing an instrument as a child can alter the brain and neural pathways in a beneficial way and the long term [94,95]. Evidence for their claims has been published [96,97,98,99,100] and was also reinforced by our interviewees (see Table A1, subtheme 1, theme 3, and subtheme 3, theme 4), despite some degree of published controversy [101]. Many interviewees stated that music could influence personal well-being and mood. Furthermore, our interviewees described that listening to a song or composition that one had listened to in the past might bring back memories that were created during that time of life (see Table A1, subthemes 1 to 5, theme 3). Music and memory are another recurring and important research topic in the scientific literature [102,103,104], especially regarding individuals with dementia [105,106].

Perhaps, as a result of their personal experience, and consistent with Barenboim’s musical kindergarten project in Berlin, Germany [95], many interviewees emphasized the need to implement active music making at different educational levels, such as in primary school courses, making it available to everyone starting from a very young age. Participants made the following statements: I think that children should be introduced to music much earlier. […] Children today are overwhelmed with the world. The methods of education go in many directions. You can’t teach everything that is necessary these days. I can imagine that music can calm children down and get them down, at least (Table A1, subtheme 3, theme 4).

An interesting aspect and additional subtheme of theme 3 was the perception of music as a way of self-conditioning. This process was explained through different statements made by several of our interviewees. The participants expressed that music could be helpful in a broad range of tasks, from simplifying household chores to making one more focused and concentrated during manual-practical tasks (see Table A1, subtheme 4, theme 3). This subtheme of self-conditioning is somewhat related to reports in the literature describing the benefits that surgeons may gain from adopting the practice and performance strategies of expert musicians [107,108]. Notably, this subtheme shows some overlap with subtheme 3 from theme 1.

Additionally, within theme 3, interviewees emphasized that active singing had particular importance. Since singing outside of a professional setting is possible without needing to learn to play an instrument, it had a special role during the interview sessions [109,110,111,112]. Everyone has a voice, and can therefore use it for oneself and or with others. Melodies that are stuck in people’s minds throughout the day may influence their way of thinking (see Table A1, subtheme 5, theme 3). Furthermore, it was stated that music could achieve stress reduction through the same pattern as pain relief, as mentioned above (see Table A1, subtheme 4, theme 4). “There is stress relief via the rhythm and establishing a relationship with oneself”. Listening to and performing music may shift one’s thoughts, take one away from a one-sided thinking pattern, and even help a person develop more stable mental health [113,114]. A person can merely focus on the sound of music, thereby putting other thoughts in the background for an amount of time, such that one may have different ideas on a subject afterwards [115].

### 4.4. Theme 4: Music for Society

Many interviewees stated that music could facilitate the building of relationships within society in various possible ways (see Table A1 and subthemes 1 and 2, theme 4). Music was described as embodying a sense of belonging, a community that can generate resources, harmony of a society. [One should make it] accessible to all population groups. It connects many people with each other, on a beautiful emotional level, not dependent on one’s origin, income, occupation (…) It is an important bridge (this citation is not included Table A1). This perception is in line with previous reports that include the notion that music plays a role in societal cohesion [109,116,117,118,119]. Interviewees also explained that music is a product of culture: It has something to do with down-to-earthiness, anchoring, and identity (see Table A1, subtheme 2, theme 4). On the other hand, music has also been perceived as constitutive of people’s youth and being associated with the evolution of social groups during young adolescence [109].

We also identified some criticism related to topics in theme 4. One such criticism concerned concert tours and traveling, which are typically a part of musicians’ schedules: I practice a beautiful art and delight people worldwide in concert halls, but what happens? Entire orchestras are flown across the globe. What happens to the waste I am creating? (See Table A1, subtheme 5, theme 4). Additionally, interviewees voiced concerns that music, especially learning to perform, is not affordable for everybody, although it is perceived as such an important factor for human development: For myself, I will go with the direction of social justice. What happens in society? Why are there so many refugees? There’s so much social injustice. People with no privilege, homeless shelters. People don’t have a place to live. We should think about community music therapy. This is the concept. Music therapy that works in refugee camps. That is something that needs to be established long-term. (Table A1, subtheme 5, theme 4).

One interviewee also stated that people can listen to music via headphones as a means of dissociating from their surroundings, such as in public transport. This perception is concordant with the idea that music can be seen as a tool of self-identification and distancing through different music genres experienced by the same individual [120].

## 5. Conclusions

A literature search conducted in January 2023 using the PubMed online database yielded approximately 6000 citations on the topic of “music and medicine”. However, to the best of our knowledge, a comprehensive, systematic description of the overall relationship between music and medicine has not yet been established. Despite the wealth of clinical research suggesting the evident effects of music on health, there remains a need to explore the underlying mechanisms that contribute to these effects.

In our qualitative study examining the perceptions of music and medicine, we found it easier to identify themes related to the impact of music on medicine (specifically in music therapy and music medicine) compared to the impact of medicine on music. Recognizing that the health status of performers and composers is intertwined with the music they create, we aimed to elucidate the interdependence between music and medicine. We sought the perspectives of attendees and invited speakers at a music and medicine conference, exploring not only what music could do for medicine but also what medicine could do for music.

The responses we received covered a wide range of perspectives, with interviewees overwhelmingly acknowledging the positive impact of music on individual development and society as a whole. They emphasized the urgent need for increased awareness and recognition of the synergistic relationship between music and medicine, suggesting that community campaigns could play a vital role in achieving this goal.

One important limitation of our study is the potential selection bias resulting from conducting interviews at the “Science and Sounds” conference. The pre-selected sample of participants may not be fully representative of the general population, potentially influencing the findings and generalizability of the study. To mitigate this bias, we made efforts to interview a diverse range of individuals across different ages and specializations.

Furthermore, although our recruitment goal was aimed at achieving data saturation and diverse perspectives, careful consideration must be given to potential selection bias. Future work is necessary to further explore these perceptions and their implications. We believe that the insights gathered from this study provide valuable information for funders and researchers, emphasizing the importance of increasing public visibility of their work to improve the lives of musicians and non-musicians, particularly those with health issues but also those without.

## Figures and Tables

**Table 1 ijerph-20-06386-t001:** Demographics and personal characteristics.

Characteristic		N	%
Number of interviewed individuals	26	100	
Country of residence			
	Germany	23	89
	Switzerland	2	8
	Italy	1	4
Urban Residence			
	Yes	24	92
	No	2	8
	Not asked		
Self-declared gender			
	Men	6	23
	Women	18	69
	Not asked/not answered	2	8
Ethnicity			
	Caucasian	22	84
	Non-Caucasian	4	15
Native language			
	German	20	77
	Not German	6	23
Age in years			
	≥60	4	15
	51–60	3	12
	41–50	5	19
	31–40	3	12
	21–30	10	38
	<21		
	Not asked/not answered	1	4
Number of children			
	4	3	12
	3	1	4
	2	4	15
	1	2	8
	0	12	46
	Not asked/not answered	4	15
Employment status			
	Full time	14	54
	Part time	6	23
	Not answered (also including students who did not explicitly state they were working)	6	23
Highest Degree			
	University	18	69
	High School	5	19
	Other	1	4
	Not asked/not answered	2	8
Job			
	Medical profession	2	8
	Performing musician (earning money with music making)	2	8
	Both medical profession and musician	3	12
	Teaching music therapy	4	15
	Music therapist	6	23
	None of the above	3	12
	Student or in training	6	23
	Not answered	2	8
Time actively spent making music			
	>10 h per week	6	23
	2–10 h per week	13	50
	<2 h per week	3	12
	None	4	15
Time consuming music			
	>40 h per week	3	12
	10–40 h per week	10	38
	2–10 h per week	13	50
	<2 h per week		
Age of first encounter listening to tonal music			
	>5 years	1	4
	2–5 years	7	27
	≤2 years	12	46
	Not answered	6	23
Age of first encounter listening to atonal music			
	>5 years	21	81
	2–5 years	1	4
	≤2 years		
	Not answered	4	15
Age of first active music making			
	>5 years	14	54
	2–5 years	5	19
	≤2 years	0	
	Not asked	4	15
	No active music making	3	12

## Data Availability

Data is given in Appendix A; further data is unavailable due to privacy or ethical restrictions.

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
