# Peer review of "Exploring the Synergy of Music and Medicine in Healthcare: Expert Insights into the Curative and Societal Role of the Relationship between Music and Medicine"

_ijerph, 2023, doi:10.3390/ijerph20146386_

Round 1

Reviewer 1 Report

Dear authors,

many thanks for the opportunity to read your interesting study. I found your design and approach very challenging for me since I am used to other forms of research. Your approach is in some instances subjective but also descriptive which is why I find it OK but also difficult to provide comments to help you improving your manuscript.

So, I just have minor comments. See blow

Line:

Line 179: This sentence can be omitted. There is no need to convince someone from his expertise. This is already shown.

 „It was led by author UK, who is an experienced qualitative researcher, and has been previously published [38].

Descriptives – lines 192 – 199 – please give reasons for choosing this collection of participants, why students, therapists etc. I would have avoided to include students – professionals may have larger knowledge etcc.. So please give reasons for this

Language:

I am not a native speaker however, there might be some incorrect words…eg.

Line 203 “For example, participants voiced” – is voiced the right word?

English flow was fine to me and I had no difficulties in reading.

Author Response

Thank you for your consideration, please see the attachment for the answers. 

Reviewer 2 Report

Thank you for the opportunity to review this interesting paper on conference attendees' opinions on the role of music in healthcare and performing arts medicine. Its subject is timely and important. I congratulate the authors on their interest in music's relationship and applicability to their professions. There are however, a number of issues that need the authors' attention that would make for a stronger paper.

The paper would strongly benefit from a careful revision of the clarity of concepts and expression of those concepts in the text. Please see my comments in the "English quality" section of this review.

Please consider modifying the title. I do not see anything in the survey or themes that addresses the "preventative" use of music and medicine. Are the opinions actually on the "relationship between science and sound" or on the role of music in medical care/what experiences participants had with performing arts medicine?

Introduction: It is an interesting idea to consider perception depending on a performer's state of health, however listing 'unhealthy' and 'healthy' performers does nothing to contribute to development of that idea. Please consider removing the lists.

Materials and methods: I am not sure of the justification of the conference details being included in this section. While interesting, I don't see how the list of presentations figures here. Please consider removing the list and including more about actual materials and methods used in this study.

Investigators: I am not accustomed to seeing the credentials and professions of investigators, especially those involved in conducting interviews and collecting data, included in materials and methods sections. Please include a description of its relevance to the study or remove them.

Interview setting: please describe when and why group interviews were used, and what effect it may have had on the answers.

Interview guide: Please explain the "flexible use of the outline" and how and why that depended on the interviewees' responses. Please explain how bias was avoided in this situation. 

In the actual interview guide, please explain the relevance of asking interviewees about their personal limitations in life.

Recruitment goal... Please explain why 30 participants was the goal and why  researchers  looked for "data saturation" Please explain how this process may have affected bias.

Data analysis: please describe why the consolidated criteria for reporting QR was used.

Results: You need to change "musical therapist" to "music therapist" 
The use of "subthemes" and "categories" creates some confusion. There are 4 categories of inquiry correct? It might be clearer to refer to the broader ideas that emerged as "themes" instead of calling them 'categories' and then using 'subthemes' (as you have) to identify specifics.  Please chose terms to make this clearer to the reader throughout the paper.

Figure 1: I don't see what contribution this figure makes. The points the authors make in the description of this figure are obvious and have been expressed in many other papers. I strongly suggest removing it. In any case including the statement about Apollo seems totally out of place.

Discussion: the first sentence state that you are presenting the results of the interviews. I think there needs to be some restructuring here. I suggest reviewing this very large section, and placing results in the "Results" section of the paper, and having the actual discussion of the results in the discussion section.

Conclusions: I am not sure that Hugo's quote is a comment on the effects of music continuing to be mysterious, but rather on music being able to express things not easily expressed with words (such as emotions) Perhaps a different quote more related to the authors' idea would make for a stronger beginning to this Conclusions section. Also, I am not sure that many would agree with the statement such as it is. It is not so much that the "effects" which at this point given the amount of clinical research many music and health professionals and neuroscientists  would consider to be evident, but perhaps that the mechanisms that lead to the effects are yet to be completely discovered.  

There are some small issues with English in the paper, though importantly the meaning of the first sentence of the Abstract "we hypothesized..." is unclear. There are also a number of terms throughout the paper whose meaning is less then clear. 

For instance "medicine in music" in English is difficult grasp. It appears that the authors are in reality referring to medical treatment for musician health and wellbeing - which is "performing arts medicine" Consider using that term to identify category 2 throughout the paper. 

Another example is in the introduction: "medicine can penetrate music via the health perception on behalf of the listener" Unfortunately the meaning of this sentence is  opaque.

Also consider " conference on music medicine, musicians’ health, and musical therapy, 21 with the aim of analyzing the interactions between both fields" There are 3 fields mentioned while both refers to 2. Do you mean music and medicine?. Do you really mean Interactions?. or relationship, mutual inclusivity, compatibility? 

Personally, I have no doubts in M&M and MT's importance, but my opinion is that the statement "our results confirm the importance of music therapy..to society as a whole" is too broad and not quite accurate. It actually confirms a cohort's opinion that it is important. Strongly consider changing this statement.

The paper would strongly benefit from a careful revision of the clarity of the text.

Author Response

Dear Reviewer,

We appreciate the time and effort you invested in reviewing our manuscript. We value your feedback and would like to thank you for your constructive comments. Your insights have been instrumental in identifying areas where our manuscript can be strengthened.

First and foremost, we agree with your observation regarding the clarity of concepts and the expression of those concepts in the text. We understand the importance of ensuring a clear and concise presentation of our findings. We have also carefully reviewed your comments in the "English quality" section. Furthermore, we appreciate your suggestion to modify the title to accordingly.

Based on your suggestions, we made necessary revisions to improve the clarity and precision of the manuscript, regarding the introduction, materials and methods, interview guide, recruitment goal, data analysis, results, discussion, and conclusions. We have furthermore agree that Figure 1 did not contribute to the overall discussion. We appreciate your feedback on the structure of the discussion section and agree that a reorganization was also needed. Finally, as mentioned above, we appreciate your feedback on the quality of the English language throughout the manuscript.

We genuinely appreciate the time and expertise you have dedicated to reviewing our manuscript. We remain open to any further comments or suggestions you may have.

We would also like to thank you for acknowledging the significance of the subject matter. We share your enthusiasm for exploring the relationship between music and healthcare, and we appreciate your recognition of our interest in this interdisciplinary field and your contribution to this study.

Juliane Hennenberg, Henriette Löffler-Stastka

Reviewer 3 Report

I would like to congratulate the co-authors for having conducted these interviews and hope that the publication of this paper will eventually be successful. Until then I have a few thoughts / comments to make:

The connection between  music and medicine was to be explored in this series of semi-structured interviews. Interviews were conducted at the Science and Sounds conference in Hamburg last year. Thus, the sample is pre-selected but includes experts in the subject. Please mention and discuss the issue of bias in your discussion section. Also, opinions of the general population would have also been interesting to explore (limitations of the study).

Why was the semi-structured interview guide not developed as usual by arranging and interviewing within focus groups? (method section)

The results from the n=26 interviews should be presented in the result section, not in the discussion. Interpretation of the various results then belongs into the discussion section.

Author Response

Dear Reviewer,

We sincerely appreciate your and encouragement for our work. We are grateful for your feedback and would like to address the points you raised.

Regarding pre-selected sample size for this study, consisting of experts in the subject matter. In the discussion section of our manuscript, we will explicitly mention and discuss the potential bias that may arise from this sample selection. We recognize the importance of addressing this limitation. We clarified and provided a rationale for our choice of conducting both individual and group interviews in the method section of the manuscript as well as another possible limitation.

We furthermore understand the importance of clearly delineating between the presentation of results and the interpretation of those results. We revised the manuscript accordingly, ensuring that the results section provides a concise and objective overview of the findings, while the discussion section offers a comprehensive analysis and interpretation.

We genuinely appreciate the time and expertise you have dedicated to reviewing our paper. We remain open to any further comments or suggestions you may have.

Once again, we thank you for your valuable feedback and suggestions.

Juliane Hennenberg, Henriette Löffler-Stastka

Round 2

Reviewer 2 Report

Thank you for considering and implementing my suggestions. I find the article much improved in clarity and flow. It provides an interesting perspective on a cross-section of professionals' beliefs and awareness of various aspects of music & medicine. Warm regards.